# L-Cysteine Synthase Enhanced Sulfide Biotransformation in Subtropical Marine Mangrove Sediments as Revealed by Metagenomics Analysis

**Shuming Mo [1,†], Jinhui Li [1,†], Bin Li [2], Muhammad Kashif [1], Shiqing Nie [1], Jianping Liao [3], Guijiao Su [1], Qiong Jiang [1], Bing Yan [2,\*] and Chengjian Jiang [1,2,\*]**

1   State Key Laboratory for Conservation and Utilization of Subtropical Agro-Bioresources, Guangxi Research Center for Microbial and Enzyme Engineering Technology, College of Life Science and Technology, Guangxi University, Nanning 530004, China; moshuming1@126.com (S.M.); ijinhui@foxmail.com (J.L.); Kashif_microbiologist@yahoo.com (M.K.); nshiqing@126.com (S.N.); sugj1972@126.com (G.S.); pandajq@sina.com (Q.J.)
2   Guangxi Key Lab of Mangrove Conservation and Utilization, Guangxi Mangrove Research Center, Guangxi Academy of Sciences, Beihai 536000, China; lihshang1@163.com
3   School of Computer and Information Engineering, Nanning Normal University, Nanning 530299, China; ljp021916@163.com
\*   Correspondence: gxybing@tom.com (B.Y.); jiangcj0520@vip.163.com (C.J.)
†   These authors contributed equally to this work.

**Abstract:** High sulfides concentrations can be poisonous to environment because of anthropogenic waste production or natural occurrences. How to elucidate the biological transformation mechanisms of sulfide pollutants in the subtropical marine mangrove ecosystem has gained increased interest. Thus, in the present study, the sulfide biotransformation in subtropical mangroves ecosystem was accurately evaluated using metagenomic sequencing and quantitative polymerase chain reaction analysis. Most abundant genes were related to the organic sulfur transformation. Furthermore, an ecological model of sulfide conversion was constructed. Total phosphorus was the dominant environmental factor that drove the sulfur cycle and microbial communities. We compared mangrove and non-mangrove soils and found that the former enhanced metabolism that was related to sulfate reduction when compared to the latter. Total organic carbon, total organic nitrogen, iron, and available sulfur were the key environmental factors that effectively influenced the dissimilatory sulfate reduction. The taxonomic assignment of dissimilatory sulfate-reducing genes revealed that *Desulfobacterales* and *Chromatiales* were mainly responsible for sulfate reduction. *Chromatiales* were most sensitive to environmental factors. The high abundance of *cysE* and *cysK* could contribute to the coping of the microbial community with the toxic sulfide produced by *Desulfobacterales*. Collectively, these findings provided a theoretical basis for the mechanism of the sulfur cycle in subtropical mangrove ecosystems.

**Keywords:** sulfate-reduction gene families; subtropical mangrove sediment; sulfide; metagenomics; L-cysteine synthase

## 1. Introduction

During human activities, including the mariculture industry [1] and farming [2], sulfides excessively accumulate and can be poisonous to the environment. In addition, the sulfide production and emission cause problems of corrosion and malodor [3]. Under anaerobic conditions, sulfides reach up to 20 mM in marine mangrove sediments [4]. Many methods, such as using chemical for the oxidation of sulfide by nitrate [5] and algal [6] methods, deal with sulfide toxicity, and the elucidation of the mechanism of microorganisms in the transformation of sulfide pollutants has gained increased interest. The dissimilatory sulfate reduction, which results in the conversion of sulfate into HS$^-$

or H$_2$S, is an important reaction in the sulfur cycle [7]. The study of the dissimilatory sulfate reduction can reveal the occurrence of all dissimilatory sulfate-reducing genes in a community. However, the sulfate reduction, a common occurrence, lacks a complete pathway in single strains [8]. The high occurrence of this phenomenon implies that, as a tightly coupled pathway by sulfate-reducing bacteria (SRB), sulfate reduction is inadequate, and environmental conditions can affect microorganisms. The dissimilatory sulfate reduction is primarily driven by SRB, and the complete absence of oxygen or low-oxygen condition (<15 µM O$_2$) is vital for SRB to gain energy [9,10]. Thus, the relationship among key environmental factors, microorganisms, and sulfate reduction in the special mangrove ecosystem should be unraveled.

The mangrove ecosystem is usually characterized as anoxic, with high levels of sulfur and salt and rich in nutrients [11]. The dissimilatory sulfate reduction drives the formation of enormous quantities of reduced sulfide. H$_2$S, a malodorous substance, can cause death in many organisms [12] and is a considerable inhibitor of anaerobic bacteria in the biological treatment of molasses wastewater. Gene families, including adenosine phosphosulfate reductase (*sat*), adenylyl sulfate reductase (*aprA/B*), and dissimilatory sulfite reductase (*dsrA/B/C*), are involved in the canonical dissimilatory sulfate-reduction pathway [13,14]. Recently, some marker genes have been applied to study the diversity of sulfur-related microorganisms [13]. The study of sulfide conversion in mangroves has gained interest. Although the diversity of the SRB has been elucidated, an understanding of sulfate reduction in these ecosystems remains insufficient [14]. Culturable microbial sulfate reduction via genomic analysis is observed in hypersaline lake [15] but is not well studied in mangrove ecosystems. The relationship between the sulfate reduction and the microbial genotype involved in this process in mangroves is also poorly understood. Furthermore, the environmental conditions that select dissimilatory sulfate-reducing gene families for frequent reliance on the sulfate reduction remain unclear.

Previous studies usually used traditional approaches (e.g., cultivation and denaturing gradient gel electrophoresis) to analyze the biochemical cycle. The polymerase chain reaction (PCR) is a technique used to make numerous copies of a specific segment of DNA quickly and accurately. However, PCR usually produces bias, resulting in inaccurate experimental results because of the lack of perfect working primers for many of the gene families involved [16]. Interestingly, metagenomics provides the opportunity to recover underexplored, rare populations and identify difficult-to-elucidate biochemical pathways [17]. However, some limitations in metagenomics analysis exist. For example, sufficient and high-quality DNA samples are essential for metagenomics [18].

In the present study, we hypothesize that the sulfide biotransformation in mangrove sediments will show unique features as a consequence of adapting to environmental conditions, and the mangrove sediments and non-mangrove sediments of differences are significant enough to drive localized changes in sulfur genes occurrence. The higher diversity and bioavailability of nutrients (i.e., NH$_4^+$, NO$_3^-$, TOC, TN, and TP) alter the microbial community and create distinct metabolic profiles. Then, the shotgun metagenomic sequencing and quantitative PCR are used to elucidate the sulfide biotransformation in the special mangrove ecosystem. A newly published database, namely SCycDB [19], is used to generate functional profiles with samples from the subtropical marine mangrove ecosystem of Beibu Gulf in China. This study aims at (1) investigating all genes involved in sulfur cycling, (2) revealing the model of sulfide biotransformation in the subtropical marine mangrove ecosystem, (3) confirming the key microorganisms involved in sulfur cycling, and (4) unravelling the effect of key environmental factors on sulfate reduction.

## 2. Materials and Methods

### 2.1. Sampling Sites and Sediment Collection

The subtropical ShanKou mangrove sediments in Beihai City, China (21°29′25.74″ N, 109°45′49.43″ E), were selected as sampling location (Figure 1). The National Shankou Natural Reserve of Mangrove is located in Guangxi Zhuang Autonomous Region, with a

coastline of 50 km and a total area of 8000 km$^2$. This has a typical structure and large areas of well-preserved natural mangroves. Samples were collected from two areas (Figure 1), i.e., mangrove sediments (MS) area covered by *Rhizophora stylosa* and non-mangrove sediments (NMS) area without any vegetation at 100 m away from the edge of the mangrove. Three samples from MS were used as rhizosphere samples (RS). RS were collected from *R. stylosa* (near the root within 3 cm). Three non-RS (NRS) were collected 1.5 m away from the three RS that had no rhizosphere. All sediment samples (0–10 cm) were collected in March 2019. Sterile polyvinyl chloride tubes and sterile bags were used to collect samples within an area of 5 m × 5 m. All samples were placed in a box filled with ice and immediately transported to the laboratory for DNA extraction and chemical analysis. RS from *R. stylosa* (near the root within 3 cm) were designated as RS1, RS2, and RS3. NRS from *R. stylosa* (1.5 m away from these roots) were labeled as NRS1, NRS2, and NRS3. NMS was assigned as NMS1, NMS2, and NMS3. RS1, RS2, RS3, NRS1, NRS2, and NRS3 were used as the group of MS.

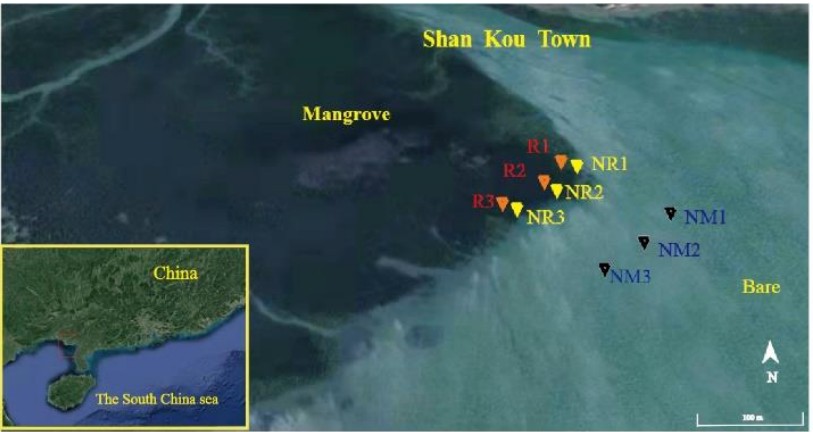

**Figure 1.** Geographic distribution of sampling sites in the subtropical mangrove ecosystem of Beibu Gulf in China. Sediment samples were collected from the National Shankou Natural Reserve of Mangrove in Beihai City (Site: 21°29′25.74″ N, 109°45′49.43″ E). Blue labels NMS1, NMS2, and NMS3 are the sampling sites of the non-mangrove sediments area (NMS). Red labels RS1, RS2, and RS3 are the sampling sites of the rhizosphere sediments area (RS); and yellow labels NRS1, NRS2, and NRS3 are the sampling sites of the non-rhizosphere area (NRS). Mangrove sampling sites include rhizosphere and non-rhizosphere sampling sites. RS1, RS2, RS3, NRS1, NRS2, and NRS3 were used as the group of MS.

Several sediment properties, such as total organic carbon (TOC), total organic nitrogen (TN), nutrients ($NH_4^+$ and $NO_3^-$), and available sulfur (AS) contents, were determined. The redox potential (ORP) was measured in situ by using a portable ORP meter (BPH-220, Bell, China). The sediment suspension was obtained through centrifugation to determine pH by using a pH meter (PHS-2C, Sanxin, China) and salinity by using a salinity meter (PAL-06S, Atago, Japan). The AS was determined using the barium sulfate turbidimetry method [20]. The total phosphorus (TP) content was determined using the alkali digestion method through the Dionex ICS-2500 Reagent-Free Ion Chromatograph (Dionex Corp., Sunnyvale, CA, USA) [21]. The iron concentration was determined using atomic absorption spectrophotometry (GBC932, Varian, Australia). The sulfide concentration was determined through spectrophotometry by using methylene blue [22]. TN and TOC were determined using an automatic carbon and nitrogen analyzer (TOC-TN 1200 Thermo Euroglas). $NH_4^+$ and $NO_3^-$ were measured using the SmartChem (Westco Scientific Instruments Inc., Brookfield, CT, USA), following the previous study [23].

### 2.2. DNA Extraction and High-Throughput Sequencing

DNA was extracted from sediments by using the FastDNA SPIN kit for soil (MP Biomedicals, USA) in accordance with the manufacturer's instructions. An at least 6-μg DNA sample was submitted to the Novogene Company (Beijing, China) for sequencing on the Illumina platform. The data output from each DNA sample was over 10 Gb.

### 2.3. Shotgun Metagenomic Sequence Processing and Analysis

Initial quality assurance/quality control, including trimming sequencing adapters and bar codes from sequence reads, was performed. Adapter sequences were removed using the SeqPrep (v1.33, https://github.com/jstjohn/SeqPrep, October 2020). Moreover, sequences < 100 bp, sequences with quality < 20, and reads containing an N base were removed using the Sickle (v1.2). Finally, clean reads were created. Clean reads were merged and assembled using the megahit (v1.1.3) with default parameters [24]. The production of the gene catalog (Unigenes) was described in a previous study [17]. Clean reads were mapped onto their assembled initial gene catalog by using the SoapAligner [25], and the number of reads in the gene alignment in all samples was calculated. For normalized abundance, unigenes were calculated on the basis of the number of reads and gene length [26].

For functional annotation, unigenes were aligned against the SCycDB database. The BLAST software of the SCycDB database was the DIAMOND (v0.9.14) [27], with parameters set to an *e*-value cutoff of $1 \times 10^{-5}$ by using the BLASTP. The results of SCycDB database output were converted into the m8 blast format. Best hits were extracted for the sulfur-cycle gene profiling. Gene families of the dissimilatory sulfate reduction were screened out. A correlation heat map was used to visualize the composition of the dissimilatory sulfate reduction across all nine samples. The Spearman correlation coefficients of abiotic factors and dissimilatory sulfate-reducing genes were calculated using the SPSS [28]. Welch's *t*-test was used for comparison of sulfur genes between the two groups.

The key gene sequences were extracted from the unigenes sequences for further taxonomy annotation. For the taxonomic annotation, unigenes and key gene sequences were aligned to the NR database (coverage > 50% and *e*-value < $1 \times 10^{-10}$) via BLASTP of DIAMOND (v0.9.14). Then, taxonomic classification of the BLASTP result was performed by using the LCA algorithm of the MEGAN software [29]. The taxonomic relative abundance was calculated based on the sum-sequencing depth of genes with same taxonomic assignment in the total depth of this gene as described in the previous study [30]. Permutational Student's *t*-test was used for comparison of microbial between the two groups.

### 2.4. Quantification of Dissimilatory Sulfite Reductase (dsrB) and Adenylyl Sulfate Reductase (aprA) Gene Copy Numbers

Quantitative polymerase chain reaction (qPCR) was performed to quantify the abundance of bacterial 16S rRNA gene and the gene coding for *dsrB* and *aprA*. qPCR was conducted using the fluorescent dye SYBR–Green approach on the Roche LightCycler® 480 II. 16S rRNA, *dsrB*, and *aprA* were quantified with primer sets 341f–797r [31], DSRp2060f–DSR4r [32], and AprA-1-FW–AprA-5-RV [33], respectively. Details on the construction of the standard plasmid were described in a previous study [34].

## 3. Results

### 3.1. Abundance and Diversity of Sulfur (sub)Gene Families

A total of 150 different sulfur gene (sub)families were annotated. The sulfur gene (sub)families in each sample ranged from 138 (RS2 sample) to 143 (RS1 sample, Supplementary Table S1). The abundance of pathways showed that the organic sulfur transformation pathway in these samples was the highest, followed by sulfur oxidation and dissimilatory sulfate reduction. The pathway with the lowest abundance was sulfur reduction (Figure 2A). As shown in Supplementary Table S1, the top 10 abundant genes were heterodisulfide reductase (*hdrA/D*), arylsulfatase (*atsA*), dimethylsulfoniopropionate

demethylase (*dmdB/A*), adenylylsulfate kinase (*cysC*), sulfur carrier protein (*tusA*), cysteine biosynthesis protein (*cysE/K*), and tetrathionate reductase (*ttrB*). Certain gene (sub)families, such as ATP sulfurylase (*aps*), sulfite dehydrogenase (*sorT*), and adenylyl sulfate reductase (*APR*), were rarely detected at the current sampling depth (Figure 2B, Supplementary Table S1). This finding indicated the low abundance of these gene (sub)families in natural environments, and deeper sequencing depths should be used in shotgun metagenomes to capture these genes.

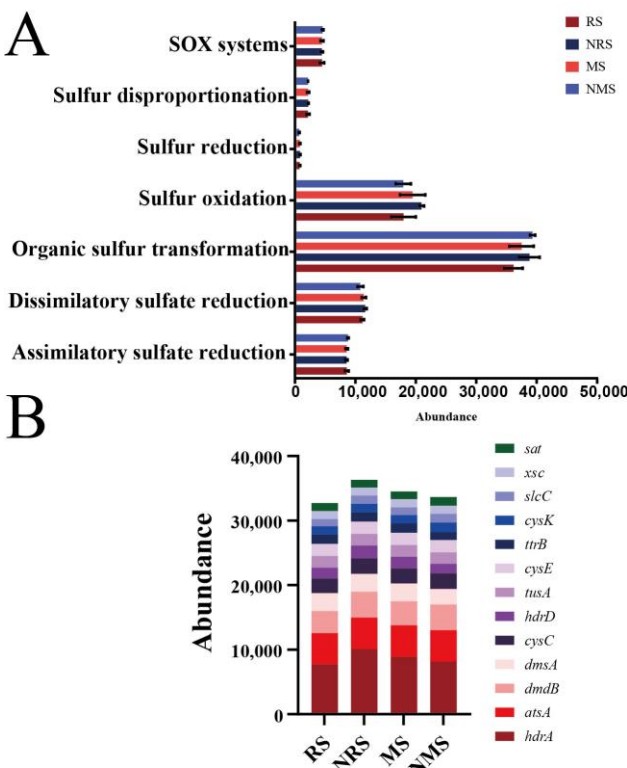

**Figure 2.** (**A**) Pathway abundance values in the samples. (**B**) Bar chart indicating the relative abundances values of 12 abundant sulfur gene (sub)families in each sample.

## 3.2. Microbial Diversity Based on Metagenomics

To study the distribution of the dissimilatory sulfate reduction in microbial communities, we annotated the taxonomy. Taxonomic assignments indicated that members of *Desulfobacterales*, which had compositions ranging from 16% to 22% across each sample, were dominant (Figure 3A). Other orders, such as *Spirochaetales*, *Cellvibrionales*, and *Gemmatimonadales*, comprised approximately 4% (Figure 3A). The abundance values of *Desulfatibacillum*, *Desulfobacterium*, and *Desulfosarcina* in MS exceeded those in NMS, whereas *Desulfobacter*, *Desulfobulbus*, and *Desulfurivibrio* in RS exceeded those in NRS (Figure 3B). A significantly enriched ($p < 0.05$) microbial taxa were also found in these samples. For example, *Betaproteobacteria*, *Methanobacteria*, and *Bacilli* were highly enriched in MS, whereas *Alphaproteobacteria*, *Thermoleophilia*, and *Bacteroidia* were abundant in NMS (Figure 3C). At the order level, microbial taxa also showed distinct distribution patterns in different regions. For example, *Desulfurobacteriales*, *Bacillales*, *Acidobacteriales*, *Nitrososphaerales*, *Clostridiales*, and *Burkholderiales* were significantly enriched in MS, whereas *Rhodobacterales*, *Flavobacteriales*, *Pseudomonadales*, *Alteromonadales*, *Cellvibrionales*, *Rhizobiales*, and *Chromatiales* were significantly enriched in NMS (Figure 3D).

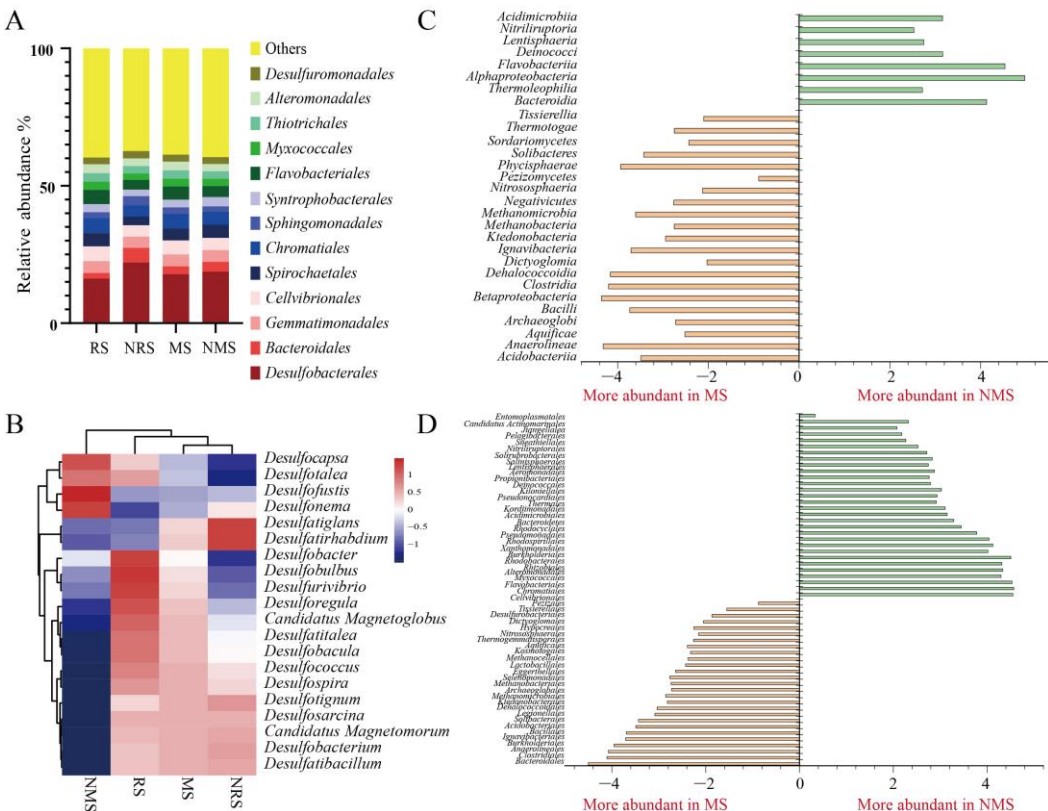

**Figure 3.** Comparison of taxonomic assignments from metagenomes. (**A**) Thirteen dominant orders with their relative abundance values (remaining orders indicated as "Others"). (**B**) Heat map of the *Desulfobacterales* order. Differential analyses between mangrove and non-mangrove regions at the (**C**) class and (**D**) order levels. For the differential analysis, the significance feature was selected using the Student's *t*-test of *p*-values, and values are the base-10 logarithm of the abundance. Positive and negative values indicate taxa that are abundant in mangroves and non-mangroves, respectively.

### 3.3. Genes for the Dissimilatory Sulfate Reduction

The prevalence of dissimilatory sulfate-reducing genes at the sites was initially analyzed by considering the complete dissimilatory sulfate-reduction pathway. Accordingly, genes encoding adenosine phosphosulfate reductase (i.e., *sat*), adenylyl sulfate reductase (i.e., *aprA/B*), and dissimilatory sulfite reductase (i.e., *dsrA/B*) were obtained (Figure 4A). In MS and NMS groups, *dsrA*, *aprA*, and sulfite reduction-associated complex DsrMKJOP multiheme protein (*dsrM/P*) were significantly high in MS (*p* < 0.05, Figure 4B). Genes encoding the sulfate adenylyltransferase subunit 2 (*cysD*) were significantly low in MS (*p* < 0.05, Figure 4B). The *dsrA* catalyzes the reduction of sulfite into sulfide, which is the terminal oxidation reaction in the sulfate respiration [35]. The *aprA* catalyzes the reduction of adenosine 5′-phosphosulfate into sulfite and AMP reversibly during the dissimilatory sulfate reduction [33]. In RS and NRS groups, genes encoding anaerobic sulfite reductase (i.e., *asrB*) and DsrC–disulfide reductase (i.e., *dsrK*) were significantly low in RS (*p* < 0.05, Figure 4C). *Sat* and *aprA* genes were more frequent than *dsrA/B* (Figure 4A). The *sat* gene accounted for 9.6–13.3% of the total dissimilatory sulfate-reducing genes in all samples (Figure 4A).

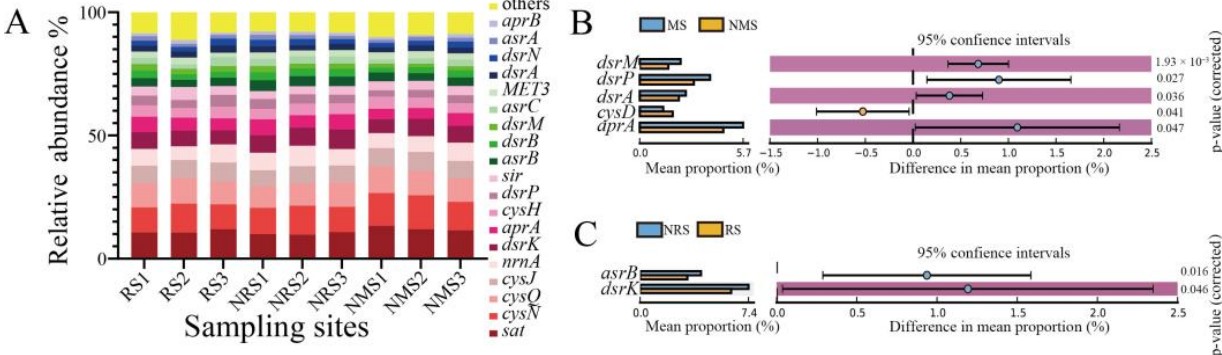

**Figure 4.** (**A**) Bar chart of the relative abundance values of 20 abundant dissimilatory sulfate-reducing gene (sub)families in each sample. Significant differences in sulfur gene (sub)families (**B**) between mangrove and non-mangrove sediments and (**C**) between rhizosphere and non-rhizosphere sediments.

### 3.4. Taxonomic Assignment of Dissimilatory Sulfate-Reducing Genes

The order *Desulfobacterales* was predominant in the microbial community (Supplementary Figure S1). Therefore, taxonomic assignments were conducted at the order level to obtain a better understanding of sulfate reduction (Table 1). The numbers of dissimilatory sulfate-reducing genes from *Desulfobacterales* and *Chromatiales* exceeded those from other taxonomic assignments, whereas remaining genes were assigned to *Rhizobiales*, *Desulfovibrionales*, *Desulfuromonadales*, and *Cellvibrionales* (Supplementary Figure S1). This finding was consistent with the microbial diversity analysis and showed that these samples were as prevalent as *Desulfobacterales* (phylum *Proteobacteria*, Figure 3A). The high abundance of *aprA* was obtained from *Desulfobacterales* and *Chromatiales* (Table 1). In addition, *asr* and *cys* gene families were obtained from *Desulfobacterales* and *Chromatiales* (Table 1). The taxonomic classification of *dsrB* was assigned to *Chromatiales* and *Desulfobacterales* in MS (11.14% and 67.26%, respectively) and NMS (22.66% and 48.50%, respectively; Supplementary Figure S3). However, the taxonomic classification of *dsrB* was assigned to *Chromatiales* and *Desulfobacterales* in RS (9.10% and 65.72%, respectively) and NRS (12.44% and 68.23%, respectively; Supplementary Figure S4).

**Table 1.** Taxonomic assignments of dissimilatory sulfate-reduction genes.

| Taxonomy | RS | NRS | NMS |
|---|---|---|---|
| Chromatiales | *cysJ; dsrK; aprA; cysN; dsrA; sir; sat; aprM; cysQ; cysNC; cysD; cysH; dsrO; asrB; dsrN; dsrP; asrA; nrnA; MET3; dsrB; asrC; HINT4; cysI* | *cysN; aprA; dsrB; dsrO; sir; sat; cysQ; dsrK; cysNC; cysD; MET3; dsrM; cysH; cysJ; asrB; dsrP; dsrN; cysI; dsrA; aprB; nrnA; dsrC; HINT4* | *cysJ; cysN; asrB; cysD; cysNC; dsrK; asrA; sat; cysI; cysH; cysQ; aprA; dsrC; sir; aprM; nrnA; dsrP; MET3; dsrM; aprB; dsrB; dsrA* |
| Desulfobacterales | *cysJ;cysN; aprA; asrB; cysQ; asrC; dsrK; nrnA; sat; dsrM; asrA; dsrN; cysH; dsrB; sir; dsrP; cysD; SAL; MET3; cysNC; dsrA; aprB; cysI; dsrO; MET22; dsrC;dsrJ* | *dsrK; cysJ; sat; aprA; cysN; cysQ; nrnA; dsrJ; dsrM; dsrC; asrB; cysNC; dsrN; cysH; dsrB; asrC; dsrA; sir; dsrP; aprB; SAL; asrA; MET3; MET22; dsrO; cysI; cysD* | *dsrK; cysJ; sat; asrB; cysN; asrA; cysQ; aprA; dsrB; MET22; nrnA; sir; dsrP; aprB; dsrJ; dsrN; dsrM; cysD; cysNC; dsrA; asrC;cysH; dsrO; cysI; MET3; dsrC* |
| Rhizobiales | *sat;dsrO; cysI; cysJ; cysH; asrA; cysQ; cysN; sir; nrnA; asrB* | *sat; aprA; cysJ; cysH; cysQ; asrB; HINT4; cysN* | *aprM; sat; dsrA; aprA; cysN; cysI; cysQ; asrB; cysJ; dsrO; sir; dsrK; nrnA* |
| Desulfovibrionales | *sat; asrC; dsrK; sir; cysN; cysJ* | *aprA; sat; dsrJ; cysN; asrC; dsrK; cysJ* | *sat; dsrK; cysN* |
| Desulfuromonadales | *cysI; cysN; asrB; nrnA; cysQ; sir; asrC* | *cysN; sir; asrC* | *cysI; cysN; asrB; sat; dsrA; sir; cysQ; dsrP; nrnA* |

### 3.5. Quantification of dsrB and aprA Genes

qPCR results showed that the abundance values of *dsrB* and *aprA* genes were $13$–$53.80 \times 10^7$ and $48$–$160.48 \times 10^7$ copies per g soil, respectively. The abundance of the 16S rRNA gene was $7$–$44 \times 10^8$ copies per g soil. The copy numbers of *dsrB* and *aprA* in MS were higher than those in NMS (Table 2). The copy numbers of *dsrB* and *aprA* in NRS were higher than those in RS (Table 2).

**Table 2.** Quantitative polymerase chain reaction results of subtropical mangrove sediments in the Beibu Gulf.

| Sample | RS | NRS | MS | NMS |
|---|---|---|---|---|
| 16S rDNA ($10^8$ copies/g soil) | $24.56 \pm 8.59$ | $36.50 \pm 4.95$ | $30.53 \pm 9.21$ | $11.02 \pm 3.34$ |
| *dsrB* ($10^7$ copies/g soil) | $54.45 \pm 10.68$ | $53.80 \pm 31.69$ | $48.54 \pm 24.22$ | $13.08 \pm 8.91$ |
| *aprA* ($10^7$ copies/g soil) | $142.54 \pm 33.07$ | $160.48 \pm 88.53$ | $151.51 \pm 67.42$ | $48.87 \pm 31.60$ |
| *dsrB* ($10^{-1}$ copies/16r rDNA) | $1.82 \pm 0.19$ | $1.43 \pm 0.78$ | $1.63 \pm 0.60$ | $1.09 \pm 0.46$ |
| *aprA* ($10^{-1}$ copies/16r rDNA) | $6.04 \pm 0.68$ | $4.31 \pm 2.21$ | $5.17 \pm 1.85$ | $4.13 \pm 1.62$ |

All results are reported as means $\pm$ standard deviation. Except the MS group, which has six samples, all groups have three replicates.

### 3.6. Sediment Properties

The correlation heat map was generated to determine the environmental factors that likely shaped the structure and the composition of dissimilatory sulfate-reducing genes and microorganisms in the mangrove sediments (Figure 5A,B). Certain sediment properties, such as pH, TOC, AS, ORP, $NH_4^+$, $NO_3^-$, TN content, TP content, iron content, salinity, and sulfide content, were determined (Supplementary Figure S2). The concentrations of AS, iron, TOC, and TN in MS were significantly higher than those in NMS ($p < 0.05$). Moreover, the concentrations of TOC, TN, and TP in RS were significantly higher than those in NRS (Supplementary Tables S3 and S4, $p < 0.05$).

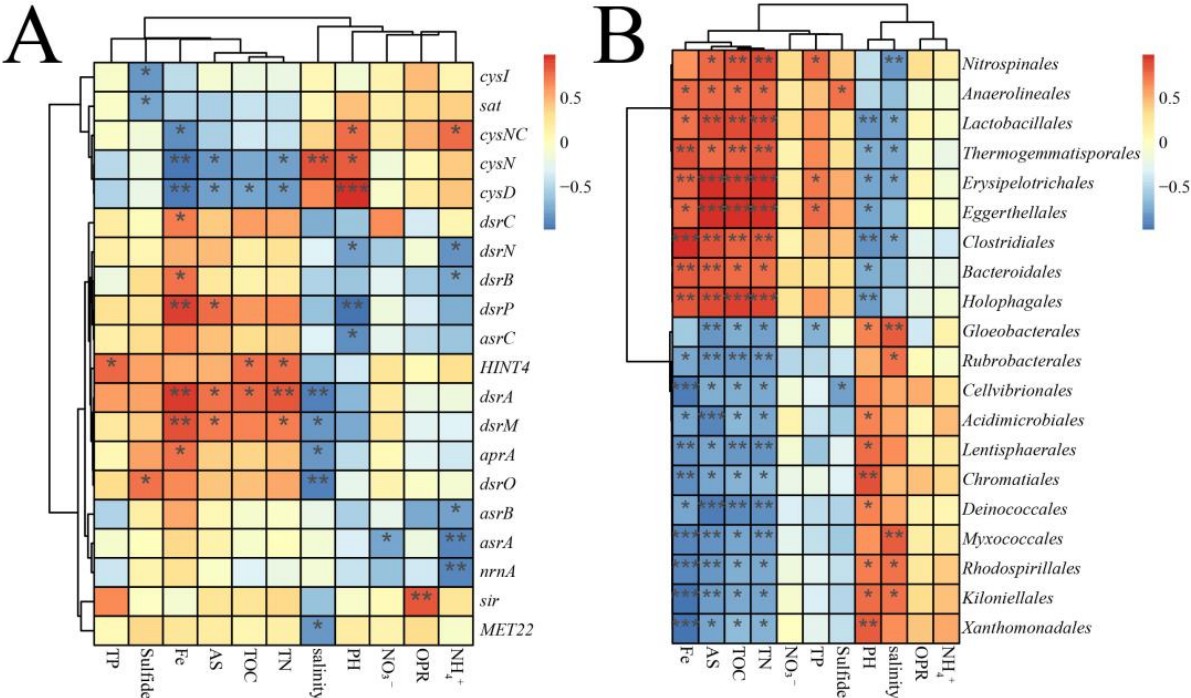

**Figure 5.** (**A**) Correlation heat map according to the *z*-scores of the 20 most abundant dissimilatory sulfate-reducing gene (sub)families with significant correlation among sediment properties. (**B**) Correlation heat map according to the *z*-scores of the 20 most abundant microbial communities with significant correlation among sediment properties. * $p < 0.05$, ** $p < 0.01$, *** $p < 0.001$.

Results showed significant correlations between sediment properties and dissimilatory sulfate-reducing gene (sub)families ($p < 0.01$, Figure 5A, Supplementary Table S2). pH was significantly correlated with *cysD*, *dsrP*, and *asrC*. The salinity content was significantly correlated with *aprA* and *dsrB/M* ($p < 0.05$). The iron content was highly correlated with *dsrA/B/C/M* and *aprA*, whereas the AS content was significantly correlated with *dsrA/M/P* ($p < 0.05$). TOC, TN, AS, Fe, salinity, and pH were significantly correlated with the microbial community, and *Chromatiales* was significantly correlated with TOC, TN, AS, and Fe ($p < 0.05$, Figure 5B).

## 4. Discussion

### 4.1. Sulfur-Cycling Genes in the Mangrove Ecosystem

In the mangrove ecosystem, the organic sulfur transformation was abundant, and 7 of the top 10 genes belonged to this pathway (Figure 2B). This finding was consistent with those of other ecosystem surveys [19], showing that organic sulfur might be the largest pool of sulfur in the mangrove ecosystem [36] and providing energy for microorganisms. The highest abundance of sulfur genes in the mangrove sediment was *hdrA*, and *Acidithiobacillus ferrooxidans* has shown this function in previous study [37]. These microorganisms link organic sulfur to inorganic metabolism [19], which can provide energy for the sulfur-cycle process. L-cysteine synthase genes (i.e., *cysK* and *cysE*) were in the top 10 sulfur genes (Figure 2B). These genes are required for the biosynthesis of L-cysteine from sulfate [38]. In addition to the organic sulfur transformation genes, sulfate adenylyltransferase genes involved in sulfate reduction were highly abundant in the top 13 genes. Therefore, the sulfide biotransformation was active in mangrove ecosystem. Given the insufficient sequencing depth or environmental factors, some sulfur genes had low abundance. In addition, sulfate reduction may be inhibited by the availability of organic matter and the concentration of sulfate [39]. TOC and AS had significantly negative correlation with the sulfate adenylyltransferase (*cysN*) in the present study ($p < 0.05$, Supplementary Table S2). The sampling depth might be insufficient because the sulfate-reduction pathway also occurred under anaerobic conditions.

### 4.2. Microorganisms Involved in the Sulfide Conversion

Many microbes were involved in the sulfate-reduction pathway [14,40]. In the present study, we found that 94 orders had dissimilatory sulfate-reducing genes, suggesting that the sulfate reduction was a tightly coupled pathway by SRB (Supplementary Figure S1). *Desulfobacterales* and other sulfur-metabolizing microorganisms provided numerous sulfate-reduction genes and produced a large accumulation of sulfide that poisoned the mangrove ecosystem (Table 1, Supplementary Figure S1). Similar observations that the *Desulfobacterales* of SRB commonly reduce sulfate to sulfide have been found previously [41]. In the present study, *Desulfobacterales* was the dominant cluster, and this finding was consistent with previous results described in mangrove ecosystems [42].

The absolute abundance obtained by qPCR indicated that MS could provide more sulfate-reducing genes than NMS (Table 2). The taxonomic classification of *dsrB* revealed that *Desulfobacterales* in MS were higher than that in NMS. This finding was similar to that obtained previously [39], where SRB increased due to pollutants. This gene was affiliated with *Deltaproteobacteria* and *Betaproteobacteria* in the mangrove ecosystem in a previous study [39]. Hence, the mangrove zone environment was conducive to the growth of sulfate-reducing microorganisms. *CysK, cysE*, and *sqr* were basically obtained from *Desulfobacterales*, *Pseudomonadales*, and *Sphingomonadales*, respectively (Supplementary Figure S5). These findings showed the occurrence of a unique transformation of sulfide pollutants by microbial communities in mangrove ecosystems.

### 4.3. Major Environmental Factors Affecting the Sulfate Reduction

Results showed that the environmental factors of mangrove ecosystem shaped the microbial community. Microbial communities are frequently influenced by environmental factors [43]. The marked enrichment of methanogens (e.g., *Methanobacteria*) and SRB (e.g., *Clostridiales* and *Burkholderiales*) in MS reflects the effect of environmental factors on microbial communities (Figure 3C,D). In addition, fewer *Chromatiales* were found in MS than in NMS (Figures 3D and 5B). The key environmental factor related to microbial communities and sulfur cycle were defined by partial Mantel tests. TP showed the most significant correlation with microbial communities and sulfur gene structure, and the observed values of rM were 0.527 and 0.685 with sulfur gene structures and microbial communities, respectively (Supplementary Table S5). These results indicate that the sulfur cycle is possibly intertwined with phosphorus cycles [44]. Previous studies demonstrated that pH is the most important environmental factor related to the microorganism community [45].

The diversity and bioavailability of nutrients may be the key environmental factors to influence sulfate-reducing genes [39]. Some studies showed that mangrove forests can retain large amounts of organic matter because mangroves are efficient in trapping and accumulating suspended matter during tidal inundation [46]. Mangrove ecosystems are rich in TOC [11] because a large amount of organic matter decomposes slowly in the absence of oxygen. In the present study, the average content of TOC in all samples was 7.78 mg/g (Supplementary Table S3). Metagenomic and qPCR data showed that the *aprA* gene in MS was higher than that in NMS (Figure 4B, Table 2). Such differences may have high organic-matter contents in MS that can supply enough carbon sources for the reduction of sulfate [47]. The present results also confirmed the significantly positive correlation among TOC, TP, TN, AS, and sulfate-reducing genes ($p < 0.05$). Phosphorus and iron are closely coupled to the activity of SRB [48]. Iron plays an important role for most organisms in electron-transfer reactions and prosthetic groups, such as hemes or iron–sulfur clusters [49]. The zerovalent iron ($Fe^0$) contributes to the formation of an anaerobic environment, and the iron sulfide precipitation could relieve the inhibition of sulfide to improve sulfate-reduction capacity, which is beneficial to SRB [50–52]. Salinity also exhibits a significant effect on the soil microorganism community structure [53]. The typical range of salinity in the mangrove ecosystem is 25–55 ppt [54]. Previous literature reports that salinity can impair the bioavailability of organic matter and the availability of nutrients in the mangrove ecosystem [55]. In the present study, the average content of salinity in all samples was 29.42 ppt (Supplementary Table S3). Thus, salinity might also influence the sulfate reduction by the bioavailability of organic substrates (Figure 5A). In the present study, TOC, TN, iron, and AS concentrations in mangrove zones were consistently higher than those in non-mangrove zones ($p < 0.05$, Supplementary Table S3), suggesting an excellent determination of the abundance of the dissimilatory sulfate reduction.

### 4.4. Mechanism of the Sulfide Conversion in the Mangrove Ecosystem

The dissimilatory sulfate reduction can lead to a high level of sulfides in the mangrove ecosystem [7]. In the present study, the average content of sulfide in all samples is 0.07 mg/g (Supplementary Table S3). The model for the pathway of the dissimilatory sulfate reduction in the mangrove ecosystem is shown in Figure 6. EC 2.7.7.4 (sulfate adenylyltransferase) had the highest abundance among enzymes. EC 1.8.99.2 (adenylyl-sulfate reductase) and EC 1.8.99.5 (dissimilatory sulfite reductase) in mangrove samples had higher abundance than those in non-mangrove samples (Figure 6). The qPCR results showed that the relative abundance values (per 16S rRNA) of *aprA* and *dsrB* in MS were higher than those in NMS, thereby confirming the accuracy of the shotgun sequencing analysis (Table 2). This result was consistent with that obtained by a previous study [56], i.e., an environment with high organic content was conducive to sulfate reduction. In RS, the abundance of EC 1.8.99.5 was low, whereas that of EC 1.8.1.2 (assimilatory sulfite reductase) was high (Figure 6). This finding indicated that rhizosphere microorganisms were conducive to the assimilation of sulfate to L-cysteine to mitigate sulfide pollution [57].

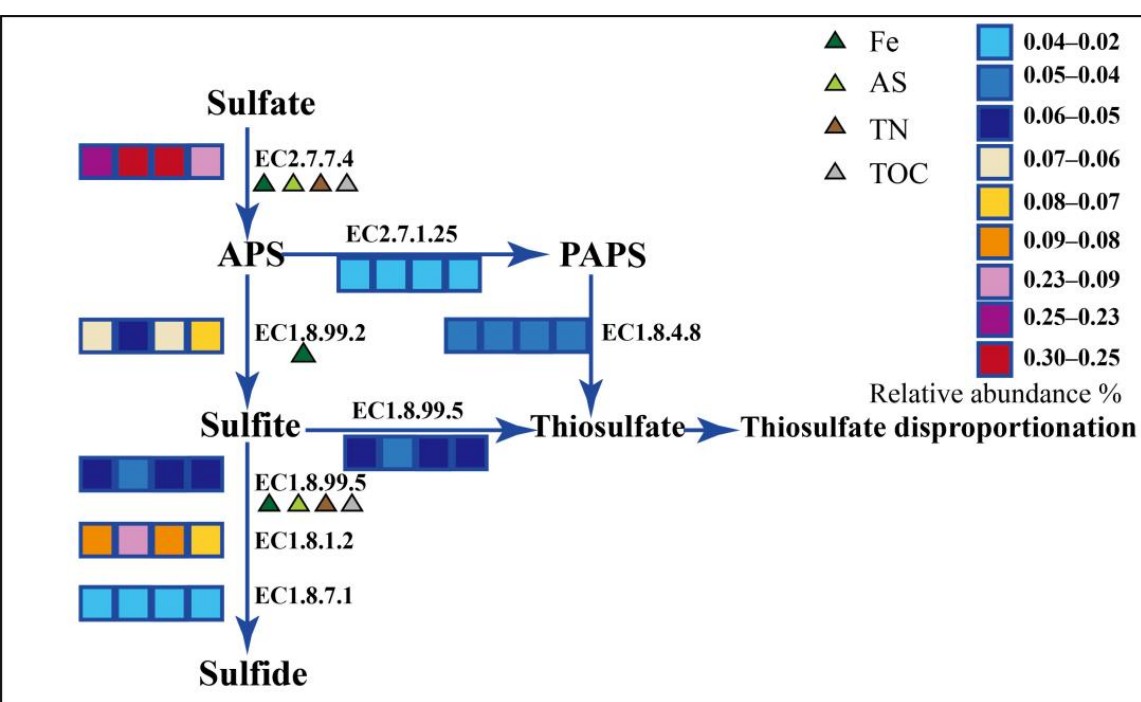

**Figure 6.** Model for the pathway of the dissimilatory sulfate reduction. Relative abundance values of gene-encoding enzymes involved in the dissimilatory sulfate reduction. The EC numbers of enzymes are boxed. The relative abundance values of enzymes in the samples are shown in the nearby color bar, in which the four segments from left to right represent MS, NMS, RS, and NRS. The effects of sediment properties on dissimilatory sulfate-reducing genes are depicted within the triangle.

The sulfide conversion in mangrove sediments included biological and abiotic processes (Figure 7A). Biogeochemical studies suggest that the sulfide metabolism is involved in chemical reaction and microbial metabolism [44]. For abiotic processes, ferric ion is buried, acts as an oxidant for sulfide in deeper sediment layers, and partly binds the produced sulfide as iron sulfide and pyrite [44]. Some studies also showed that the addition of iron can mitigate $H_2S$ [3]. In addition, the sulfide oxidation by $O_2$, $NO_3^-$, and $Fe_3^+$ as electron acceptors produces sulfur, whereas $Mn_4^+$ as electron acceptor produces thiosulfate or sulfate. For biological processes, providing the gene families of *cysK* and *cysE* can reduce sulfide levels to protect local community members [58,59]. These activities are required for the biosynthesis of L-cysteine from sulfate, which is the major way for microorganisms to assimilate environmental inorganic sulfur sources [38]. The absence of this gene can either inhibit the growth of organisms in that community or slow down their growth. Some species of *Thiobacillus* (order *Nitrosomonadales*) can use sulfide to support their growth (Supplementary Figure S1) [60]. Mangrove ecosystems could convert sulfide into L-cysteine, and this finding was consistent with that reported in mangroves in a previous study [17]. The L-cysteine biosynthesis was active in the subtropical mangrove ecosystem (Supplementary Table S1). Reports regarding this finding are few. In the present study, the polysulfide formation is another way to reduce sulfide levels by the quinone oxidoreductase (*sqr*, Supplementary Table S1). In addition, our results showed that the abundance values of *cysK* and *cysE* were higher than those of *aprA/B* and *dsrA/B* (Figure 7B). *AprA* and *dsrB* are the key genes responsible for the dissimilatory sulfate reduction [32]. These results suggested that mangroves could mitigate sulfide pollution.

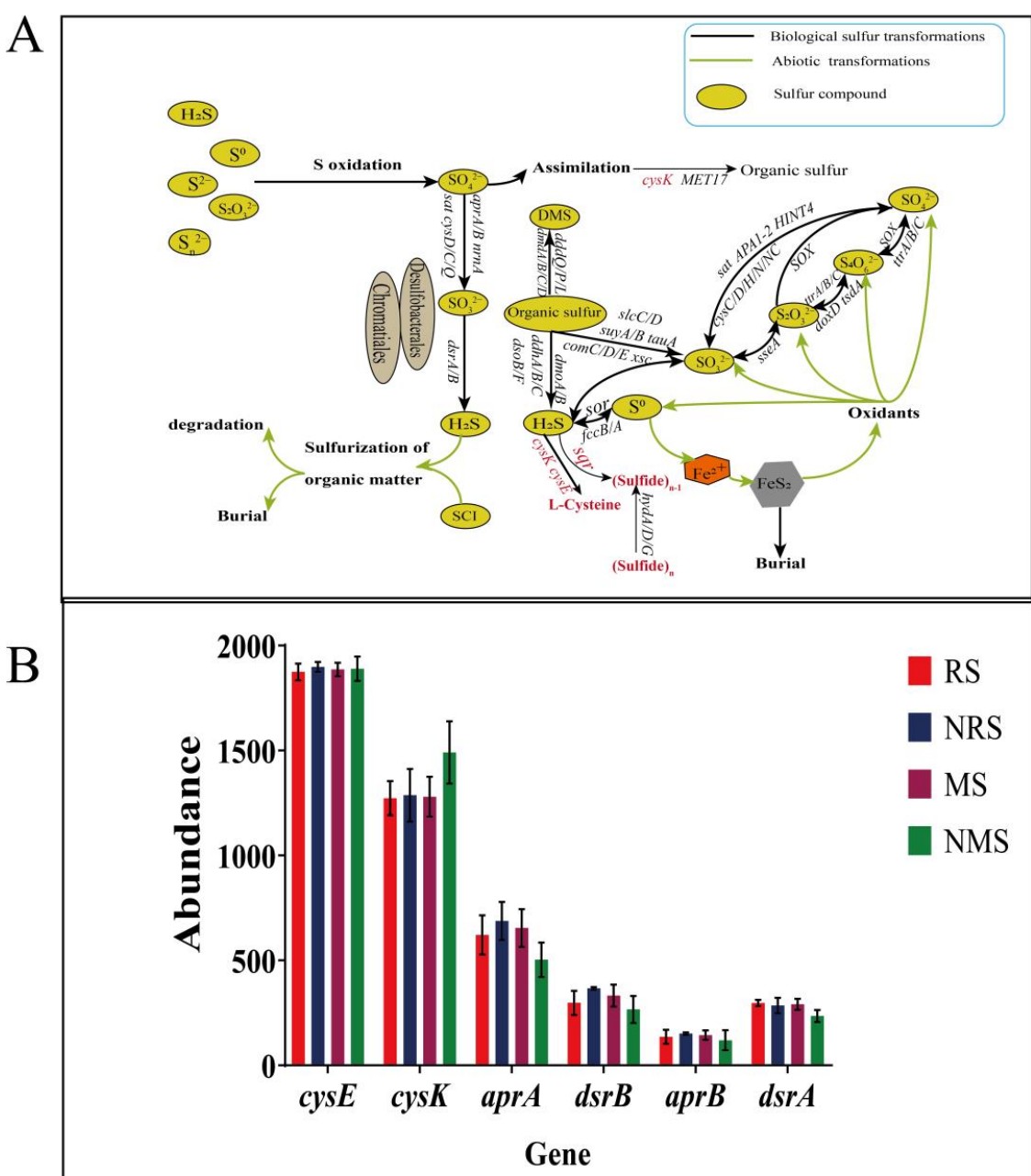

**Figure 7.** (**A**) Conceptual depiction of sulfide conversion in the mangrove ecosystem, including biological and abiotic processes. Biological processes include oxidation and reduction of sulfur compounds. Black lines depict biological sulfur transformations by microorganisms. Green lines depict abiotic-reaction-mediated sulfur transformations to pyrite ($FeS_2$). Sulfur compounds are depicted within yellow eclipses. (**B**) Abundance values of *cysK, cysE, dsrA, dsrB, aprA*, and *aprB* in samples.

## 5. Conclusions

This study demonstrated that the pathway of organic sulfur transformation was the most dynamic activity in the subtropical marine mangrove ecosystem. TP was the dominant environmental factor that drove the sulfur cycle and microbial communities. Sulfur bacteria, especially *Desulfobacterales*, are the primary executor of sulfide biotransformation. The concentrations of AS, iron, TOC, and TN in mangrove soils were significantly higher than those in non-mangrove soils. These environmental factors in mangrove soils enhanced the metabolism of sulfate reduction. *Desulfobacterales* and *Chromatiales* were found to be responsible for the dissimilatory sulfate reduction. Furthermore, *Chromatiales* were most

sensitive to environmental factors. The cysteine synthase could contribute to biotransformation of sulfide. Mangrove sediment microbiomes assimilated sulfide into L-cysteine to mitigate sulfide pollution. This study provided a theoretical basis for the sulfur-cycle mechanism in subtropical mangrove wetland ecosystems.

**Supplementary Materials:** The following are available online at https://www.mdpi.com/article/10.3390/w13213053/s1, Figure S1: The 20 dominant dissimilatory sulfate-reduction genes taxonomy order level are shown with their relative abundances; Figure S2: Sediment properties. (A) TOC (mg/kg), TN (mg/kg), and TP (mg/kg). (B) Contents (mg/kg) of AS and sulfide. (C) pH, salinity, and ORP. (D) Contents (mg/g) of $NH_4^+$, $NO_3^-$, and Fe; Figure S3: Taxonomic classification of key functional genes retrieved from the samples. (A) The key genes enriched in the MS, (B) The key genes enriched in the NMS; Figure S4: Taxonomic classification of key functional genes retrieved from the samples. (A) The key genes enriched in the RS, (B) The key genes enriched in the NRS; Figure S5: Taxonomic classification of key functional genes retrieved from the samples. (A) *cysK*, (B) *cysE*, (C) *sqr*; Table S1: Functional gene abundances; Table S2: Impact of sediment properties on dissimilating sulfate-reduction gene families; Table S3: Sediment properties in mangrove samples and non-mangrove samples; Table S4: Sediment properties in rhizosphere samples and non-rhizosphere samples; Table S5: Partial Mantel test to evaluate the relative importance of sediments variables in determining microbial communities and the distribution of sulfur genes in subtropical mangrove ecosystem.

**Author Contributions:** Conceptualization, C.J. and B.Y.; methodology, S.M. and J.L. (Jinhui Li); software, S.N. and M.K.; validation, B.L., J.L. (Jianping Liao) and Q.J.; formal analysis, G.S.; investigation, B.L., C.J., and B.Y.; data curation, G.S. and J.L. (Jinhui Li); writing—original draft preparation, S.M.; writing—review and editing, C.J.; visualization, S.M.; supervision, C.J.; project administration, C.J. and S.M.; funding acquisition, C.J. and B.Y. All authors have read and agreed to the published version of the manuscript.

**Funding:** This work was supported by the National Natural Science Foundation of China (Grant No. 31760437), Science and Technology Basic Resources Investigation Program of China (Grant No. 2017FY100704), the Funding Project of Chinese Central Government Guiding to the Guangxi Local Science and Technology Development (Grant No. GUIKEZY21195021), Natural Science Fund for Distinguished Young Scholars of Guangxi Zhuang Autonomous Region of China (Grant No. 2019GXNSFFA245011), and the Innovation Project of Guangxi Graduate Education (Grant No. YCSW2021064).

**Institutional Review Board Statement:** Not applicable.

**Informed Consent Statement:** Not applicable.

**Data Availability Statement:** Metagenomic data are available at NCBI, accession numbers: PRJNA767118.

**Conflicts of Interest:** The authors declare that they have no known competing financial interests or personal relationships that could have appeared to influence the work reported in this paper.

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
