# Peer review of "L-Cysteine Synthase Enhanced Sulfide Biotransformation in Subtropical Marine Mangrove Sediments as Revealed by Metagenomics Analysis"

_water, doi:10.3390/w13213053_

Round 1

Reviewer 1 Report

The manuscript entitled "L-Cysteine synthase enhanced sulfide biotransformation in subtropical marine mangrove sediments as revealed by meta-genomics analysis" is highlighting the scope of journal. The research work is very valuable. The topic is very interesting and critical for the field. The results lead to interesting conclusions  Therefore, I recommend this manuscript for publication in Water journal after minor revision.

In order to improve of manuscript quality I suggest you consider following comments:

General comments:

Please use to italic for the name of the bacterial taxa.

In some part of the manuscript you use 16S notation and other you use 16s. Please check and unify.

Why is section 6 "Patents" of the manuscript blank ?

Materials and Methods:

More details on the 16S metadata analysis should be presented, e.g. alph-diversity calculations and species classification.

Line: 139-140. This information is not necessary here. It's already listed in Data Availability Statement.

I hope you will find my comments useful.

Reviewer 2 Report

water-1433250

L-Cysteine synthase enhanced sulfide biotransformation in subtropical marine mangrove sediments as revealed by metagenomics analysis

General comments:

The present study provides information on the mechanism of the sulfur cycle in subtropical mangrove ecosystems, using metagenomic sequencing and quantitative polymerase chain reaction analysis. The authors concluded that total phosphorus was the dominant environmental factor driving the sulfur cycle and microorganism communities and that the cysteine synthase could contribute to biotransformation of sulfide when mangrove sediment microbiomes assimilate sulfide into L-cysteine to mitigate sulfide pollution. They also provide a model for the pathway of the dissimilatory sulfate reduction in the mangrove ecosystem.

Overall, the manuscript is well written and structured, therefore I recommend its publication. However, I think the work may still be improved. Changes and suggestions are described below in the specific comments for all sections of the paper.

Specific comments:

Abstract

Page 1, Line 19: I suggest changing to: “…marine mangrove ecosystem has gained increased interest.”

Page 1, Line 19: You should emphasize in this sentence that you are already talking about your work, so I suggest something like: “Thus, in the present study, the sulfide biotransformation in subtropical mangroves ecosystem was accurately evaluated using metagenomic sequencing and quantitative polymerase chain reaction analysis.”

Page 1, Line 23: I suggest changing to: “…the sulfur cycle and microbial communities.”

Page 1, Line 23: I suggest changing to: “We compared mangrove and nonmangrove soils and found that the former enhanced metabolism that was related to sulfate reduction when compared to the latter.”

Page 1, Line 28: Correct to: “Chromatiales were most sensitive…”.

Introduction

Page 2, Line 80: I suggest changing to: “In the present study, we hypothesize…”

Page 2, Line 89: I suggest changing to: “This study aims at (1) investigating all genes involved in sulfur cycling, (2) revealing the model of sulfide biotransformation in the subtropical marine mangrove ecosystem, (3) confirming the key microorganisms involved in sulfur cycling, and (4) unravelling the effect of key environmental factors on sulfate reduction.”

Materials and Methods

Page 3, Lines 100: You should put the name of the species Rhizophora stylosa in italic, correct here and in the rest of the paragraph.

In Figure 1, you should put the correct codes of the sampling stations as it is written in the text (RS, NRS and NMS).

Page 3, Line 103: You should write “nutrients”, as you analysed two types. Also, I believe you do not need to put e.g. before the nutrients, as they were in fact the ones you have analysed.

Results

I believe the Figures (Fig. 3, 4. 6 and 7), in general, are too small. Try to increase the font of the texts.

Page 5, Line 205: Can you please say which statistical analysis did you use? You should mention it in the methods.

Page 8, Line 274: Add a comma in the sentence: “…TP content, iron content, salinity,”

Discussion

Page 10, Line 333: I suggest changing to: “These findings showed the occurrence of a unique transformation of sulfide pollutants by microbial communities in mangrove ecosystems.”

Page 10, Line 340: I suggest changing to: “In addition, fewer Chromatiales were found in MS than in NMS (Figures 3D and 5B).”

Page 10, Line 345: I suggest changing to: “These results indicate that the sulfur cycle is possibly intertwined with phosphorus cycles.”

Conclusions

Page 12, Line 432: I believe you do not need to put (p < 0.05) as you already mention it in the results.
